

# Further observations on the security of SPECK32-like ciphers using machine learning

Zezhou Hou, Jiongjiong Ren and Shaozhen Chen

Information Engineering University, Zhengzhou, Henan, China

Corresponding author
Jiongjiong Ren,
jiongjiong_fun@163.com

## ABSTRACT

With the widespread deployment of Internet of Things across various industries, the security of communications between different devices is one of the critical concerns to consider. The lightweight cryptography emerges as a specialized solution to address security requirements for resource-constrained environments. Consequently, the comprehensive security evaluation of the lightweight cryptographic primitives—from the structure of ciphers and cryptographic components—has become imperative. In this article, we focus on the security evaluation of rotation parameters in the SPECK32-like lightweight cipher family. We establish a machine learning-driven security evaluation framework for the rotational parameter selection principles—the core of SPECK32's design architecture. To assess different parameters security, we develop neural-differential distinguishers with considering of two distinct input difference models: (1) the low-Hamming-weight input differences and (2) the input differences from optimal differential characteristics. Our methodology achieves the security evaluation of 256 rotation parameters using the accuracy of neural distinguishers as the evaluation criteria. Our results illustrate the parameter (7,3) has stronger ability to resist machine learning-aided distinguishing attack compared to the standard (7,2) configuration. To our knowledge, this represents the first comprehensive study applying machine learning techniques for security assessment of SPECK32-like ciphers. Furthermore, we investigate the reason for the difference in the accuracy of neural distinguishers with different rotation parameters. Our experimental results demonstrate that the bit bias in output differences and truncated differences is the important factor affecting the accuracy of distinguishers.

# INTRODUCTION

Classic differential cryptanalysis (*Biham & Shamir, 1991*) is one of the most powerful cryptanalysis techniques used in modern block ciphers. And the core of differential cryptanalysis to succeed is to search for some high-probability differential characteristics. These high-probability differential characteristics can be referred to as differential distinguishers. In recent years, some automatic tools and dedicated heuristic search algorithms have been used to search for high-probability characteristics. The attackers transform the cryptanalysis models of search for high-probability characteristics into

Mixed Integer Linear Programming (MILP) problems (*Sun et al., 2014*; *Bagherzadeh & Ahmadian, 2020*), Constraint Programming (CP) problems (*Gérault, Minier & Solnon, 2016*; *Sun et al., 2017*), and Boolean satisfiability problem or satisfiability modulo theories (SAT/SMT) (*Kölbl, Leander & Tiessen, 2015*; *Song, Huang & Yang, 2016*), which can be handled by some appropriate solvers. The use of automatic tools and heuristic algorithms improves the ability to analyze block ciphers. However, these machine-assisted technologies do not help attackers obtain more features of block ciphers than differential characteristics.

With the development of data-driven learning and computing hardware, machine learning (ML) has made remarkable progress and is widely used in important research areas such as computer vision and speech recognition. Just as that the use of automatic tools speeds up the search for differential characteristics, combining classic cryptanalysis with deep learning to efficiently and intelligently evaluate the security of block ciphers is one of the trends of current research. A remarkable work of combining classic cryptanalysis with machine learning is shown in CRYPTO 2019. In CRYPTO 2019, *Gohr (2019)* shows that a simple neural network could be trained to be a superior cryptographic distinguisher performing a real-or-random cryptographic distinguishing task. Gohr trains the Residual Network (ResNet) (*He et al., 2016*) to capture the non-randomness of the distribution of round-reduced SPECK32/64 (*Beaulieu et al., 2015*), where the trained neural networks are known as neural distinguishers ($\mathcal{ND}_s$). As a result, $\mathcal{ND}_s$ of five-, six-, seven-round SPECK32/64 are trained to distinguish the ciphertext pairs whose corresponding plaintext differences hold $(0x40, 0x0)$ and the random ones. The obtained $\mathcal{ND}_s$ exhibit noticeable advantages over pure differential characteristics. Gohr's work brings a new direction of combining classic cryptanalysis with machine-aided methods. There are many related works built upon Gohr's work (*Hou et al., 2020*; *Benamira et al., 2021*; *Bao et al., 2022*; *Bacuieti, Batina & Picek, 2022*; *Chen et al., 2022*; *Lu et al., 2023*). Moreover, there are some researches showing the different performances using the different training configurations (*Baksi et al., 2021*, *2023*). Using the different activation functions, deep learning libraries and network architectures, there is a significant difference in the accuracy of neural distinguishers.

**Motivation:** before a new cryptography algorithm is published and used in the Internet Protocol, it is important to evaluate the security of the primitive from multiple perspectives, such as differential cryptanalysis, linear cryptanalysis, and other cryptanalysis (*Hong et al., 2006*; *Suzaki et al., 2012*; *Koo et al., 2017*). With the development of cryptanalysis theory, an important branch of cryptanalysis theory for cryptographic primitives is the investigation of the security of cryptographic components. In addition, more and more researchers are concerned about the influence of the choice of cryptographic components on the security of ciphers. A notable block cipher is SIMON (*Beaulieu et al., 2015*), whose cryptographic components attracted a lot of attention. SIMON is a lightweight block cipher family published by researchers from the National Security Agency (NSA) in 2013. And its round function only uses basic arithmetic operations such as XOR, bitwise AND, and bit rotation, which makes SIMON simple and elegant.

In CRYPTO 2015, *Kölbl, Leander & Tiessen (2015)* investigate the general class of parameters of SIMON-like ciphers and evaluate the security of different rotation parameters based on differential and linear cryptanalysis. *Kölbl, Leander & Tiessen*'s *(2015)* work opens up new directions, especially the choice and justifications of the parameters for SIMON-like ciphers. In the Applied Cryptography and Network Security conference (ACNS) 2016, *Kondo, Sasaki & Iwata (2016)* classify the strength of each rotation parameter of SIMON-like ciphers with respect to integral and impossible differential attacks. In Information Security Practice and Experience conference (ISPEC) 2016, *Zhang & Wu (2016)* investigate the security of SIMON-like ciphers against integral attacks. These investigations enrich the results related to the security of the cryptographic component of SIMON, and the NSA has not disclosed a parameter selection criterion until now. Although there is a lot of work on the parameters of SIMON-like ciphers, for SPECK, the twin of SIMON proposed by the NSA, there is little research on the parameters of SPECK. Compared with the SIMON cipher designed for optimal hardware performance, the SPECK cipher is tuned for optimal software performance. And there are a lot of differences, including the encrypt function and key schedules, which makes it necessary to analyze the choice of parameters for the SPECK cipher. In the objective case that the choice of parameters leads to different encrypt functions and further delivers the different security performance for the IoT services, considering the different attack modes comprehensively, the best parameter should be offered.

In addition, as a new tool, the neural distinguishers will help researchers obtain more information about the choice of parameters and perform a more comprehensive security assessment for cryptographic primitives. Much more research on the parameters of SPECK using neural distinguishers is essential.

**Main contribution:** in this article, we investigate both the security of SPECK32-like ciphers against neural differential cryptanalysis, as well as the design choice of NSA. We show with experiments that the original choice of rotation parameter is not one of the strongest, and then several superior candidates are recommended. To our knowledge, this is the first time to evaluate the security of SPECK32-like ciphers using neural distinguishers. In addition, we analyze the reason for the difference in accuracy using different rotation parameters. The contributions of this work are summarized as follows:

- **Train neural distinguishers using low-Hamming plaintext differences and evaluate the security of the rotation parameters.** Considering that the low-Hamming weight input difference usually leads to a better differential characteristic, we train neural distinguishers of seven-round SPECK32-like using plaintext differences with Hamming weight at most $2$[1]. The accuracy result shows that there is a huge gap in the accuracy of neural distinguishers for different rotation parameters, which ranges from 50% to 100%. Then, using the accuracy of neural distinguishers as the security evaluation criterion, we evaluate the security of 256 rotation parameters. We show that the original rotation parameter (7,2) in SPECK32 has reasonably good resistance against the distinguishing attack based on neural distinguishers, but may not be the best alternative. And the

[1] In *Baksi et al. (2023)*, the authors utilized the input differences with Hamming weight exceeding 2 to train the neural distinguishers. However, the set of such input differences possesses substantial cardinality, rendering it computationally infeasible to consider all elements. Consequently, the Hamming weight was restricted to at most 2 in our work.

SPECK32-like using (2,10) or (7,3) has a lower accuracy of the neural distinguishers than using other rotation parameters.

- **Train neural distinguishers using input differences of optimal truncated characteristics and evaluate the security of the rotation parameters.** Inspired by the work of *Benamira et al. (2021)*, it is meaningful to train neural distinguishers using input differences of optimal truncated characteristics. We first build an SAT/SMT model for searching for differential characteristics of SPECK32-like. And utilizing the automatic tool Z3-solver (*de Moura & Bjørner, 2008*), the optimal five-, six-round differential characteristics are obtained. Then we train neural distinguishers of seven-round SPECK32-like using these input differences of five-, six-round characteristics. Meanwhile, we complete the security evaluation of 256 rotation parameters. Similarly, (7,2) is not the best choice using optimal truncated characteristics. The SPECK32-like using (7,3) has a stronger ability to resist the distinguishing attack based on neural distinguishers.

- **Analyze the reason for the difference in the accuracy of neural distinguishers.** We choose five rotation parameters $\{(15,1), (1,7), (7,8), (3,12), (8,11)\}$ and the corresponding plaintext differences to train neural distinguishers of seven-round SPECK32-like. The five neural distinguishers have different accuracy, which are 97.33%, 85.55%, 76.89%, 66.67% and 54.96% respectively. We also record the bits biases in five-, six-, seven-round differences for five rotation parameters. It is found that the bit biases in output differences and truncated differences are related to the accuracy of neural distinguishers. And the more bits whose frequency is significantly different from 0.5, the higher the accuracy of neural distinguisher seems to have.

**Organisation of the article:** the remaining of this article is organized as follows. "Preliminaries" gives the notations and a brief description of SPECK and introduces the neural distinguishers and the distinguishing attack. "Evaluate the Security of Different Rotation Parameters using Neural Distinguishers" introduces the security assessment of SPECK32-like. "Analysis of the Reason for the Difference in Accuracy" researches the reason for the difference in the accuracy of neural distinguishers. "Conclusion and Future Work" gives conclusions and future work.

## PRELIMINARIES

### Notations
The notations used in this work are shown in Table 1.

### A brief description of SPECK
SPECK (*Beaulieu et al., 2015*) is a family of lightweight block ciphers proposed by the National Security Agency (NSA). The aim of SPECK is to fill the need for secure, flexible, and analyzable lightweight block ciphers. It is a family of lightweight block ciphers with block sizes of 32, 48, 64, 96, and 128 bits. Table 2 makes explicit all the parameter choices for all versions of SPECK proposed by NSA. As shown in Fig. 1, for $k_i \in \mathbb{F}_2^n$, the SPECK2n

**Table 1 Some notations of this article.**

| Notation | Description |
|---|---|
| $\oplus$ | Bitwise XOR |
| $+$ | Addition modulo $2^n$ |
| $S^j$ | Left circular shifts by j bits |
| $S^{-j}$ | Right circular shifts by j bits |
| $k_i$ | $i$-round subkey $k_i = k_i^{n-1}||\ldots||k_i^0$ |
| $\mathbb{F}_2$ | The finite field consisting of the two elements $0, 1$ |
| $\mathbb{F}_2^n$ | The $n$-dimensional vector space over $\mathbb{F}_2$ |
| $\mathscr{HW}(\Delta)$ | Hamming weight of |
| $\mathscr{ND}_t$ | $t$-round neural distinguisher |
| SPECK32$_{(\alpha,\beta)}$ | SPECK-like cipher with a block size of 32 bits and using $(\alpha, \beta)$ as the rotation parameter |

**Table 2 SPECK parameters.**

| Block size $2n$ | Key size | Rot $\alpha$ | Rot $\beta$ | Rounds $T$ |
|---|---|---|---|---|
| 32 | 64 | 7 | 2 | 22 |
| 48 | 72 | 8 | 3 | 22 |
| | 96 | 8 | 3 | 23 |
| 64 | 96 | 8 | 3 | 26 |
| | 128 | 8 | 3 | 27 |
| 96 | 96 | 8 | 3 | 28 |
| | 144 | 8 | 3 | 29 |
| 128 | 128 | 8 | 3 | 32 |
| | 192 | 8 | 3 | 33 |
| | 256 | 8 | 3 | 34 |

round function is the map $R_{k_i}:\mathbb{F}_2^n \times \mathbb{F}_2^n \to \mathbb{F}_2^n \times \mathbb{F}_2^n$ defined by Eq. (1), where $\alpha$ and $\beta$ are the rotation parameters. As it is beyond our scope, we refer to *Beaulieu et al. (2015)* for the description of the key schedule.

$$R_{k_i}(x_i, y_i) = \left((S^{-\alpha}x_i + y_i) \oplus k_i, S^\beta y_i \oplus (S^{-\alpha}x_i + y_i) \oplus k_i\right). \tag{1}$$

In this article, we are interested not only in the original SPECK parameters but also in investigating the entire design space of the SPECK-like function. In addition, only SPECK-like ciphers with a block size of 32 bits are used in this article. In the rest of this article, we denote by SPECK32$_{(\alpha,\beta)}$ the variant of SPECK-like cipher with a block size of 32 bits, where the round function uses $(\alpha, \beta)$ as the rotation parameter.

## Overview of neural distinguishers

Given a fixed plaintext difference $\Delta$ and a plaintext pair $(P, P')$, the resulting ciphertext pair $(C, C')$ is regarded as s sample. Each sample will be attached with the label $Y$, which is shown in Eq. (2).

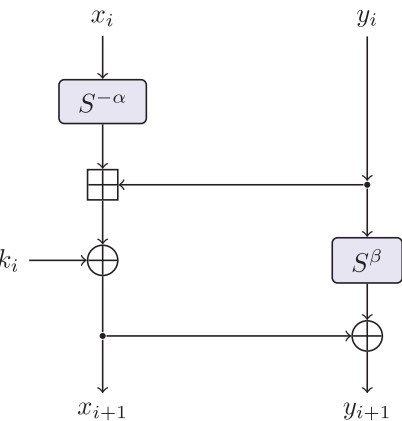

**Figure 1 The round function of speck.**

$$Y = \begin{cases} 1, & if \ P \oplus P' = \Delta \\ 0, & else \end{cases}.$$ (2)

A neural network is trained over enough samples labeled 1 and 0. In addition, half of the training data come from ciphertext pairs labeled 1, and the rest comes from ciphertext pairs labeled 0. For these samples with label 1, their ciphertext pairs are from a specific distribution related to the fixed input difference. For these samples with label 0, their ciphertext pairs are from a uniform distribution due to their random input differences. If a neural network can obtain a stable distinguishing accuracy higher than 50% in the test set, we call the trained neural network a neural distinguisher ($\mathcal{ND}$). *Gohr (2019)* chooses deep residual neural networks (*He et al., 2016*) to train neural distinguishers and obtains effective neural distinguishers of five-round, six-round and seven-round SPECK32$_{(7,2)}$.

*Gohr (2019)* explains the reason for choosing $\Delta = (0x40, 0x0)$ as the plaintext difference: it transitioned deterministically to the low-Hamming weight difference $(0x8000, 0x8000)$. And *Benamira et al. (2021)* propose a detailed analysis of the inherent workings of Gohr's $\mathcal{ND}$. They show with experiments that the $\mathcal{ND}$ generally relies on the differential distribution on the ciphertext pairs, but also on the differential distribution in truncated rounds.

## Distinguishing attack

In the differential attack, it is pivotal to distinguish encryption function from a pseudo-random permutation, which is done with the help of the differential characteristic. For an $r$-round differential characteristic $\Delta_{in} \xrightarrow{2^t} \Delta_{out} (2^t > 2^{-n})$ of a target block cipher with block size $n$ bits, we calculate the output difference given the fixed input difference $\Delta_{in}$. If the ratio of the output difference to $\Delta_{out}$ is about $2^{-t}$, then we can distinguish the block cipher from a pseudo-random permutation. This is called the distinguishing attack for block ciphers.

And the $\mathcal{ND}$ can also help the adversary to distinguish encryption function from a pseudo-random permutation. Let $F: \mathbb{F}_2^n \rightarrow \mathbb{F}_2^n$ be a permutation. The $\mathcal{ND}_{target}$ is a neural distinguisher of the target block cipher and $\Delta_{target}$ is the plaintext difference used by

$\mathcal{ND}_{target}$. The attackers can obtain $N$ ciphertext pairs encrypted by the plaintext difference $\Delta_{target}$. Using the $N$ ciphertext pairs as input, the $\mathcal{ND}$ will predict their labels. If the ratio of samples labeled one exceeds 0.5, the prediction is that $F$ is not a pseudo-random permutation.

For Gohr's neural distinguishers, the accuracy of the distinguishing attack is about 60% for a seven-round SPECK32$_{(7,2)}$, if only a pair of ciphertext is used in the distinguishing attack. And the accuracy of the distinguishing attack using a pair of ciphertext is the same as the accuracy of $\mathcal{ND}$.

*Gohr (2019)* proposes the combine-response distinguishers (**CRD**) using multiple ciphertext pairs from the same distribution. The **CRD** uses Eq. (3) and neural distinguishers to achieve higher accuracy of the distinguishing attack. And the more ciphertext pairs are used by **CRD**, the higher the accuracy of the distinguishing attack. The details of **CRD** are shown in *Bao et al. (2021)*.

$$v \leftarrow \sum_{i=1}^{m} \log_2\left(\frac{v_i}{1 - v_i}\right). \tag{3}$$

In addition, it is obvious that the higher the accuracy of the neural distinguisher, the better the effect of the distinguishing attack. For a SPECK32-like cipher, it is easier to distinguish cipher data from pseudo-random data, if the neural distinguisher with higher accuracy is used, which indicates that the rotation parameter used by the SPECK32-like cipher is not good with respect to the security against neural distinguishers and distinguishing attacks.

## EVALUATE THE SECURITY OF DIFFERENT ROTATION PARAMETERS USING NEURAL DISTINGUISHERS

The designers of SPECK gave no justification for their choice of rotation parameters. Here, we compare the security of the rotation parameters using the accuracy of neural distinguishers as the criterion. We consider all rotation parameters $(\alpha, \beta)$ and check them from two kinds of plaintext differences models, where $\alpha \in \{0, \ldots, 15\}$ and $\beta \in \{0, \ldots, 15\}$. And our experiment shows that the origin parameter of SPECK32 is not the best choice. As a result of our investigation, considering the accuracy of neural distinguishers, we give a recommendation on the choice of parameters.

### Setting

**Network Architecture.** A neural network is used to train neural distinguishers of SPECK32$_{(\alpha,\beta)}$, and the neural network is similar to the one used in *Gohr (2019)*. The network comprises three main components: the input layer, the iteration layer, and the output layer, which is shown in Fig. 2. In this neural network, the input layer mainly converts the data format to make the iteration layer extract the features of data. And in the iteration layer[2], it learns the features of data from the encryption function or the pseudo-random permutation. Then the output layer converts the extracted features to output values.

[2] Note that we use five residual blocks in the iteration layer.

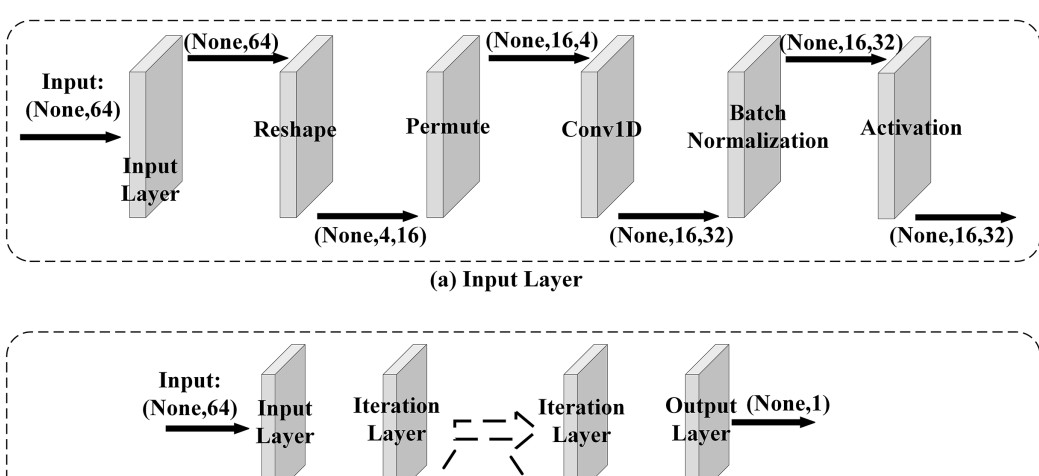

(a) Input Layer

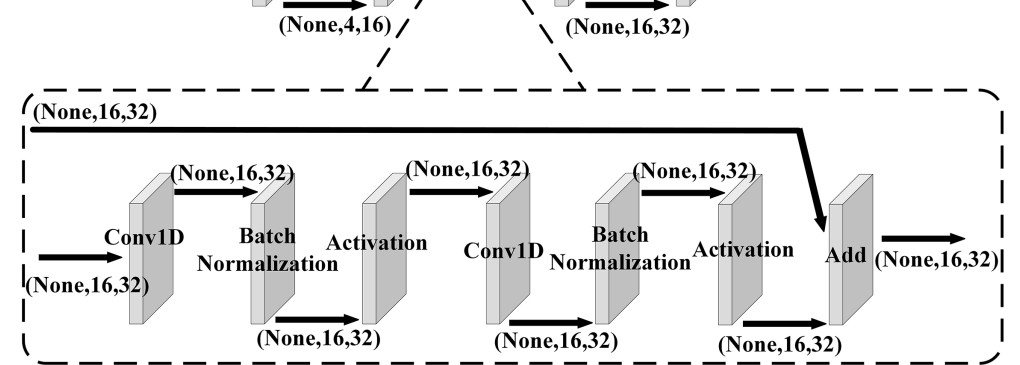

(b) Iteration Layer

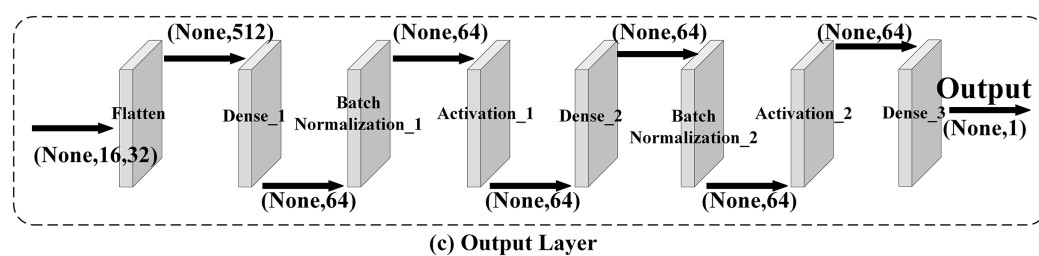

(c) Output Layer

**Figure 2  Network architecture.**

**Computing Environment.** In this article, a computing server is used to train neural distinguishers, which is equipped with Intel Xeon (R) Gold 6226R@2.90GHz*2, Nvidia GeForce RTX3090*8, 512GB RAM. The experiment is conducted by Python 3.8, cudnn 8.1, cudatoolkit 11.2 and Tensorflow 2.5 in Ubuntu20.04.

**Hyper-Parameter Setting.** The hyper-parameters used in the training neural distinguishers are shown in Table 3.

**Accuracy of $\mathcal{ND}$.** The accuracy of the neural distinguisher $\mathcal{ND}$ is calculated by Algorithm 1. In the input of Algorithm 1, the test data set is generated in the same way as the training set. Half of the test data are labeled 1, and the rest is labeled 0. The $Y_{test}$

**Table 3 List of hyper-parameters.**

| Hyper-parameters | Value |
|---|---|
| Batch size | 5,000 |
| Epochs | 15 |
| Regularization parameter | $10^{-4}$ |
| Optimizer | Adam |
| Loss function | MSE (mean-squared-error) |

---

**Algorithm 1  Calculate the accuracy of $\mathcal{ND}$.**

**Input**   Neural Distinguishers: $\mathcal{ND}$
    Test Data Set: $Data_{test}$
    Label Set: $Y_{test}$

**Output**   Accuracy of $\mathcal{ND}$: $Acc_{\mathcal{ND}}$

1: $flag \leftarrow 0$
2: **for** $sample \in Data_{test}$ **do**
3:    $Y_{sample} \leftarrow Y_{test}.sample$
4:    $Y'_{sample} \leftarrow \mathcal{ND}(sample)$
5:    **if** $Int(Y'_{sample}) == Y_{sample}$ **then**
6:      $flag \leftarrow flag + 1$
7:    **end if**
8:  **end for**
9:    $Acc_{\mathcal{ND}} \leftarrow flag/len(Data_{test})$
10:   **return** $Acc_{\mathcal{ND}}$

---

contains all the real labels of the samples in $Data_{test}$. In the Stage 3 of Algorithm 1, $Y_{sample}$ is the real label of *sample*. In the Stage 4, the $\mathcal{ND}$ will calculate the features of *sample* and return a value $Y'_{sample}$, which ranges from 0 to 1. And if $Y'_{sample} > 0.5$, then $Int(Y'_{sample}) = 1$, otherwise $Int(Y'_{sample}) = 0$. $Int(Y'_{sample})$ is the predicted label of *sample*. In the Stage 9, the size of $Data_{test}$ is denoted by $len(Data_{test})$, and the size of $Data_{test}$ is $10^7$ in this article.

## A perspective of low-Hamming weight plaintext differences

In classic differential cryptanalysis, the researchers prefer to choose the low-Hamming weight input differences to search for differential characteristics. And these automatic tools also always return the optimal differential characteristics with the low-Hamming weight input differences. For the target block cipher, using low-Hamming weight input differences is more advantageous in the number of rounds of the differential characteristics. There are existing works about neural distinguishers, and most of them choose low-Hamming weight plaintext differences to train neural distinguishers. For seven-round SPECK32$_{(\alpha,\beta)}$, all neural distinguishers are trained using the same low-Hamming weight differences. Then we save the accuracy of all neural distinguishers. The security of rotation parameters will be evaluated using the accuracy of neural

distinguishers. In this section, we limit and focus on the plaintext differences with Hamming weight at most 2.

### Training neural distinguishers

**Data Generation.** Consider a plaintext difference $\Delta$ with $\mathscr{H}\mathscr{W}(\Delta) \leqslant 2$ and the target cipher seven-round $\text{SPECK32}_{(\alpha,\beta)}$. Randomly generate $N$[3] plaintext pairs denoted by $P_{pair}$. The half of plaintext pairs are generated by using $\Delta$ as the plaintext difference. The rest of the plaintext pairs are generated using $\frac{N}{2}$ random values as the plaintext differences. Encrypt $N$ plaintext pairs using seven-round $\text{SPECK32}_{(\alpha,\beta)}$ and obtain $N$ ciphertext pairs denoted by $C_{pair}$. These ciphertext pairs are called the training data set. Each of the training data is labeled with a value 0 or 1, where 0 means the corresponding plaintext pair uses a random value as the plaintext difference, and 1 from the plaintext difference $\Delta$.

[3] Note that all neural distinguishers of $\text{SPECK32}_{(\alpha,\beta)}$ are trained with $10^7$ pairs, the same number as in *Gohr (2019)*.

**Target Cipher.** We focus on the seven-round $\text{SPECK32}_{(\alpha,\beta)}$, where $0 \leqslant \alpha \leqslant 15$ and $0 \leqslant \beta \leqslant 15$.

### Result

Considering the Hamming weight of the plaintext differences, there are 528 differences with Hamming weight at most 2. And for the seven-round $\text{SPECK32}_{(\alpha,\beta)}$, 528 neural distinguishers are trained using different plaintext differences. For all neural distinguishers of seven-round $\text{SPECK32}_{(\alpha,\beta)}$, we choose the neural distinguisher with the highest accuracy as the representative of all neural distinguishers, and use the accuracy of the representative as the accuracy of seven-round $\text{SPECK32}_{(\alpha,\beta)}$, which is shown in Algorithm 2.

Using Algorithm 2, we obtain the neural distinguisher with the highest accuracy, and choose the neural distinguisher as the representative. For seven-round $\text{SPECK32}_{(\alpha,\beta)}$, the accuracy of the representative is denoted by $Acc_{(\alpha,\beta)}$. It is obvious that the higher $Acc_{(\alpha,\beta)}$, the better the effect of the distinguishing attack, and the lower the security of $(\alpha, \beta)$. The accuracy of all representative neural distinguishers is shown in Table 4.

As shown in Table 4, the standard rotation parameter (7,2) does not seem to be always optimal if we only consider the accuracy of the neural distinguishers. And the accuracy is the lowest using (2,10) as the rotation parameter. Considering that there is a slight fluctuation in calculating all accuracy, a rotation parameter is considered good if its accuracy is less than 51%. It is found that $\text{SPECK32}$-like ciphers using (2,10) and (7,3) have better performance upon distinguishing attack with the accuracy less than 50.5%.

## A perspective of truncated differences

*Benamira et al. (2021)* propose an interpretation of Gohr's five-round $\mathscr{N}\mathscr{D}$. *Benamira et al. (2021)* explore the influence of ciphertext pairs and further find the influence of intermediate states. For five-round $\mathscr{N}\mathscr{D}$, the neural distinguisher finds the difference of certain bits at round 3 and 4. And they give a method about how to choose plaintext differences to train $r$-round neural distinguishers, that is to choose the input differences of $r-1$- or $r-2$-round optimal differential characteristics.

Inspired by *Benamira et al. (2021)*, we further investigate the security of the rotation parameters using the input differences of the optimal truncated characteristics.

---

**Algorithm 2  Obtain the representative neural distinguisher of SPECK32$_{(\alpha,\beta)}$.**

**Input** Neural Distinguishers Set of SPECK32$_{(\alpha,\beta)}$: $\mathscr{ND}_{(\alpha,\beta)}$
**Output** The representative distinguisher of SPECK32$_{(\alpha,\beta)}$: $\mathscr{ND}^{(\alpha,\beta)}$

1: $acc \leftarrow 0$
2: $\mathscr{ND}^{(\alpha,\beta)} \leftarrow \{\}$
3: **for** $\mathscr{ND} \in \mathscr{ND}_{(\alpha,\beta)}$ **do**
4:    $Acc_{\mathscr{ND}} \leftarrow$ Use Algorithm 1 to obtain the accuracy of $\mathscr{ND}$
5:    **if** $Acc_{\mathscr{ND}} > acc$ **then**
6:       $\mathscr{ND}^{(\alpha,\beta)} \leftarrow \mathscr{ND}$
7:       $acc \leftarrow Acc_{\mathscr{ND}}$
8:    **end if**
9: **end for**
10: return $\mathscr{ND}^{(\alpha,\beta)}$.

**Table 4** $Acc_{(\alpha,\beta)}$ **using low-Hamming weight difference.**

| $Acc_{(\alpha,\beta)}$ | $\beta$ | | | | | | | |
|---|---|---|---|---|---|---|---|---|
| | 0 | 1 | 2 | 3 | 4 | 5 | 6 | 7 |
| $\alpha$ | | | | | | | | |
| 0 | 100.00%[A] | 99.52%[A] | 99.51%[A] | 99.13%[A] | 97.90%[A] | 93.20%[A] | 91.84%[A] | 92.54%[A] |
| 1 | 100.00%[A] | 94.29%[A] | 81.56%[B] | 66.61%[D] | 66.14%[D] | 69.80%[D] | 68.12%[D] | 85.55%[B] |
| 2 | 99.95%[A] | 82.89%[B] | 50.55%[E] | 50.62%[E] | 50.54%[E] | 50.60%[E] | 58.86%[E] | 59.68%[E] |
| 3 | 99.61%[A] | 68.72%[D] | 50.58%[E] | 50.66%[E] | 50.58%[E] | 53.11%[E] | 50.66%[E] | 51.13%[E] |
| 4 | 93.49%[A] | 60.19%[D] | 50.63%[E] | 50.71%[E] | 55.33%[E] | 50.62%[E] | 50.57%[E] | 50.67%[E] |
| 5 | 85.31%[B] | 61.86%[D] | 50.55%[E] | 55.40%[E] | 50.63%[E] | 54.53%[E] | 60.14%[D] | 50.70%[E] |
| 6 | 84.23%[B] | 59.45%[E] | 54.43%[E] | 50.57%[E] | 50.59%[E] | 57.03%[E] | 50.53%[E] | 50.56%[E] |
| 7 | 90.58%[A] | 70.22%[C] | 60.78%[D]* | 50.49%[E] | 50.58%[E] | 50.61%[E] | 50.63%[E] | 50.65%[E] |
| 8 | 86.80%[B] | 82.34%[B] | 66.28%[D] | 55.80%[E] | 55.03%[E] | 56.57%[E] | 56.15%[E] | 65.26%[D] |
| 9 | 86.29%[B] | 62.37%[D] | 50.59%[E] | 50.53%[E] | 51.78%[E] | 50.58%[E] | 61.49%[D] | 80.97%[B] |
| 10 | 89.07%[B] | 61.13%[D] | 50.58%[E] | 50.56%[E] | 50.55%[E] | 54.34%[E] | 69.88%[D] | 63.03%[D] |
| 11 | 84.51%[B] | 57.90%[E] | 50.63%[E] | 50.53%[E] | 57.83%[E] | 66.58%[D] | 56.18%[E] | 50.61%[E] |
| 12 | 92.26%[A] | 62.62%[D] | 61.80%[D] | 63.92%[D] | 70.32%[C] | 60.22%[D] | 50.61%[E] | 50.61%[E] |
| 13 | 96.48%[A] | 77.30%[C] | 75.53%[C] | 76.90%[C] | 65.86%[D] | 50.64%[E] | 50.55%[E] | 50.50%[E] |
| 14 | 98.70%[A] | 90.61%[A] | 89.85%[B] | 83.90%[B] | 70.02%[C] | 50.58%[E] | 50.58%[E] | 50.58%[E] |
| 15 | 99.22%[A] | 97.33%[A] | 93.88%[A] | 85.95%[B] | 66.34%[D] | 57.42%[E] | 59.04%[E] | 59.86%[E] |
| $Acc_{(\alpha,\beta)}$ | $\beta$ | | | | | | | |
| | 8 | 9 | 10 | 11 | 12 | 13 | 14 | 15 |
| $\alpha$ | | | | | | | | |
| 0 | 92.40%[A] | 94.44%[A] | 92.87%[A] | 87.86%[B] | 98.03%[A] | 99.59%[A] | 99.99%[A] | 100.00%[A] |
| 1 | 74.51%[C] | 59.66%[E] | 74.18%[C] | 61.82%[D] | 89.69%[B] | 98.45%[A] | 99.97%[A] | 100.00%[A] |
| 2 | 58.39%[E] | 50.57%[E] | 50.49%[E] | 50.58%[E] | 78.10%[C] | 96.38%[A] | 99.93%[A] | 99.97%[A] |

(Continued)

| $Acc_{(\alpha,\beta)}$ | $\beta$ | | | | | | | |
|---|---|---|---|---|---|---|---|---|
| | 8 | 9 | 10 | 11 | 12 | 13 | 14 | 15 |
| 3 | 54.14%[E] | 50.66%[E] | 50.54%[E] | 50.64%[E] | 66.67%[D] | 89.17%[B] | 94.78%[A] | 99.12%[A] |
| 4 | 54.90%[E] | 50.56%[E] | 50.57%[E] | 56.41%[E] | 71.39%[C] | 66.84%[D] | 68.23%[D] | 82.38%[B] |
| 5 | 54.91%[E] | 50.55%[E] | 54.59%[E] | 67.10%[D] | 57.91%[E] | 50.51%[E] | 50.62%[E] | 62.67%[D] |
| 6 | 59.33%[E] | 59.86%[E] | 67.49%[D] | 57.23%[E] | 50.59%[E] | 50.63%[E] | 50.67%[E] | 66.65%[D] |
| 7 | 76.89%[C] | 73.38%[C] | 62.71%[D] | 50.58%[E] | 50.63%[E] | 50.51%[E] | 50.68%[E] | 57.28%[E] |
| 8 | 88.26%[B] | 83.74%[B] | 67.21%[D] | 54.96%[E] | 54.68%[E] | 56.71%[E] | 57.33%[E] | 69.90%[D] |
| 9 | 71.38%[C] | 50.56%[E] | 50.61%[E] | 50.61%[E] | 52.20%[E] | 51.36%[E] | 60.91%[D] | 81.51%[B] |
| 10 | 60.58%[D] | 50.61%[E] | 50.62%[E] | 57.76%[E] | 54.83%[E] | 50.56%[E] | 58.66%[E] | 64.57%[D] |
| 11 | 56.64%[E] | 50.58%[E] | 56.59%[E] | 54.93%[E] | 50.68%[E] | 52.54%[E] | 50.60%[E] | 64.67%[D] |
| 12 | 55.10%[E] | 50.53%[E] | 50.54%[E] | 50.54%[E] | 54.66%[E] | 50.64%[E] | 50.63%[E] | 61.50%[D] |
| 13 | 55.10%[E] | 50.58%[E] | 50.53%[E] | 53.36%[E] | 50.50%[E] | 50.67%[E] | 50.66%[E] | 59.16%[E] |
| 14 | 64.97%[D] | 54.73%[E] | 53.20%[E] | 50.59%[E] | 50.64%[E] | 50.55%[E] | 50.66%[E] | 74.60%[C] |
| 15 | 85.03%[B] | 69.15%[D] | 58.65%[E] | 61.05%[D] | 65.81%[D] | 70.96%[C] | 79.07%[C] | 91.31%[A] |

**Note:**

[*] There is the slight fluctuation in calculating accuracy. So the value is lower than in *Gohr (2019)*.

## Training neural distinguishers

Before training the neural distinguishers, we build an SAT/SMT model to search for differential characteristics of $\text{SPECK}32_{(\alpha,\beta)}$. Then we search for the exact five-, six-round differential characteristics of $\text{SPECK}32_{(\alpha,\beta)}$. The seven-round neural distinguishers of $\text{SPECK}32_{(\alpha,\beta)}$ are trained using the input difference of five-, six-round differential characteristics as the plaintext difference of neural distinguishers.

**Obtain plaintext differences.** The core of searching for differential characteristics of $\text{SPECK}32_{(\alpha,\beta)}$ is the log-time algorithm of computing differential probability of the addition shown in *Lipmaa & Moriai (2001)*. We construct the SAT/SMT model for searching for differential characteristics of $\text{SPECK}32_{(\alpha,\beta)}$. And the SAT/SMT model is suitable for the SAT/SMT solver Z3-solver (*de Moura & Bjørner, 2008*). Then we search for the exact five-, six-round differential characteristics of $\text{SPECK}32_{(\alpha,\beta)}$ with the help of Z3-solver. The Z3 solver can help judge whether there is a feasible solution to the model under the constraints, and the solver can return the feasible solutions if the model has feasible solutions. These feasible solutions are the effective differential characteristics. These input differences of five-, six-round characteristics will be used as the plaintext differences of neural distinguishers[4].

    The details of obtaining plaintext differences are shown in Algorithm 3. The input of Algorithm 3 is the SAT/SMT model for searching for $r$-round differential characteristics of $\text{SPECK}32_{(\alpha,\beta)}$. The model consists of multiple variables and their differential propagation equations. In the Stage 7, the solver will determine whether there is a feasible solution to the model under the condition that the differential characteristic is effective and the differential probability is $2^{-P}$. If there is a feasible solution, the solver will return *sat*,

[4] These input differences of five-, six-round characteristics are saved in github. com.

---

**Algorithm 3**    **Obtain plaintext differences.**

**Input** SAT/SMT model: $model^r_{(\alpha,\beta)}$
**Output** Plaintext Differences Set: $PDS$
1: $Flag \leftarrow \textbf{\textit{unsat}}$
2: $P \leftarrow -1$
3: $PDS \leftarrow \{\}$
4: $S \leftarrow \{\}$
5: **while** $Flag == \textbf{\textit{unsat}}$ **do**
6:      $P \leftarrow P + 1$
7:      $Flag \leftarrow model^r_{(\alpha,\beta)}(2^{-P})$
8: **end while**
9: **while** $Flag == \textbf{\textit{sat}}$ **do**
10:      $S \leftarrow model^r_{(\alpha,\beta)}(2^{-P}).solution$
11:      $PDS \leftarrow PDS + S.input$
12:      $model^r_{(\alpha,\beta)} \leftarrow model^r_{(\alpha,\beta)}.add(var.input \neq S.input)$
13:      $Flag \leftarrow model^r_{(\alpha,\beta)}(2^{-P})$
14: **end while**
15: return $PDS$

---

otherwise the solver will return $\textbf{\textit{unsat}}$. In Stages 5–8, exhaustive search is used to maximize the differential probability of $r$-round SPECK32$_{(\alpha,\beta)}$. In Stage 10, the solver returns the feasible solution, and the feasible solution is the exact differential characteristic of $r$-round SPECK32$_{(\alpha,\beta)}$. The input difference is saved in Plaintext Difference Set ($PDS$). In Stage 12, we add the new constraint ($var.input \neq S.input$) to the model. The $var.input$ refers to the variables associated with the input difference, and the new constraint makes the solver search for more input differences.

**Data Generation.** Consider the plaintext difference $\Delta$ obtained by Algorithm 3 and the target cipher seven-round SPECK32$_{(\alpha,\beta)}$. Randomly generate $10^7$ plaintext pairs denoted by $P_{pair}$. The half of plaintext pairs are generated by using $\Delta$ as the plaintext difference. The rest are generated using $\frac{10^7}{2}$ random values as the plaintext differences. Encrypt these plaintext pairs using the seven-round SPECK32$_{(\alpha,\beta)}$ and obtain $10^7$ ciphertext pairs denoted by $C_{pair}$. These ciphertext pairs are used to train neural distinguishers.

**Target Cipher.** We focus on the seven-round SPECK32$_{(\alpha,\beta)}$, where $0 \leqslant \alpha \leqslant 15$ and $0 \leqslant \beta \leqslant 15$.

### Result

Similar to "A Perspective of Low-Hamming Weight Plaintext Differences", for each of the rotation parameters, there are more than 1 neural distinguishers. And we choose the neural distinguisher with the highest accuracy as the representative. Utilizing these representative neural distinguishers, the security of seven-round SPECK32$_{(\alpha,\beta)}$ is evaluated. The accuracy of seven-round SPECK32-like ciphers is shown in Table 5.

**Table 5** $Acc_{(\alpha,\beta)}$ **using optimal truncated characteristic.**

| $Acc_{(\alpha,\beta)}$ | $\beta$ | | | | | | | |
|---|---|---|---|---|---|---|---|---|
| | 0 | 1 | 2 | 3 | 4 | 5 | 6 | 7 |
| $\alpha$ | | | | | | | | |
| 0 | 100.00%[A] | 99.44%[A] | 99.14%[A] | 97.85%[A] | 89.19%[B] | 77.57%[C] | 77.79%[C] | 71.96%[C] |
| 1 | 95.26%[A] | 93.28%[A] | 60.73%[D] | 59.55%[E] | 59.77%[E] | 60.08%[D] | 59.73%[E] | 85.29%[B] |
| 2 | 80.97%[B] | 67.23%[D] | 58.17%[E] | 50.44%[E] | 50.29%[E] | 53.04%[E] | 59.66%[E] | 52.07%[E] |
| 3 | 72.02%[C] | 53.42%[E] | 50.53%[E] | 52.69%[E] | 50.39%[E] | 52.91%[E] | 50.30%[E] | 50.40%[E] |
| 4 | 69.19%[D] | 52.81%[E] | 50.36%[E] | 50.34%[E] | 55.93%[E] | 50.42%[E] | 51.05%[E] | 51.60%[E] |
| 5 | 70.67%[C] | 53.34%[E] | 50.44%[E] | 55.22%[E] | 50.40%[E] | 56.52%[E] | 55.49%[E] | 50.31%[E] |
| 6 | 69.27%[D] | 53.64%[E] | 54.60%[E] | 51.57%[E] | 53.21%[E] | 53.67%[E] | 52.33%[E] | 50.35%[E] |
| 7 | 77.19%[C] | 64.82%[D] | 56.62%[E] | 50.35%[E] | 50.36%[E] | 50.31%[E] | 50.43%[E] | 53.81%[E] |
| 8 | 86.65%[B] | 82.18%[B] | 64.63%[D] | 55.36%[E] | 54.47%[E] | 54.51%[E] | 54.04%[E] | 55.30%[E] |
| 9 | 66.83%[D] | 55.52%[E] | 50.51%[E] | 50.46%[E] | 50.38%[E] | 50.46%[E] | 51.99%[E] | 70.00%[D] |
| 10 | 70.07%[C] | 52.80%[E] | 50.80%[E] | 50.31%[E] | 50.36%[E] | 53.69%[E] | 69.67%[D] | 57.09%[E] |
| 11 | 68.76%[D] | 54.29%[E] | 50.42%[E] | 50.20%[E] | 53.89%[E] | 64.56%[D] | 51.19%[E] | 50.46%[E] |
| 12 | 76.39%[C] | 53.31%[E] | 52.93%[E] | 53.26%[E] | 70.59%[C] | 55.85%[E] | 51.10%[E] | 50.32%[E] |
| 13 | 86.44%[B] | 56.18%[E] | 73.44%[C] | 74.83%[C] | 59.64%[E] | 57.25%[E] | 53.81%[E] | 50.36%[E] |
| 14 | 95.49%[A] | 90.65%[A] | 89.26%[B] | 67.42%[D] | 60.43%[D] | 51.27%[E] | 51.38%[E] | 50.30%[E] |
| 15 | 97.93%[A] | 96.26%[A] | 89.39%[B] | 73.05%[C] | 62.41%[D] | 57.26%[E] | 56.73%[E] | 56.34%[E] |
| | 8 | 9 | 10 | 11 | 12 | 13 | 14 | 15 |
| 0 | 92.44%[A] | 83.02%[B] | 79.18%[C] | 73.12%[C] | 73.28%[C] | 72.45%[C] | 75.95%[C] | 89.35%[B] |
| 1 | 65.22%[D] | 51.93%[E] | 60.44%[D] | 60.30%[D] | 62.67%[D] | 98.59%[A] | 82.92%[B] | 100.00%[A] |
| 2 | 53.79%[E] | 50.26%[E] | 51.63%[E] | 58.94%[E] | 62.33%[D] | 63.86%[D] | 84.06%[B] | 86.86%[B] |
| 3 | 53.30%[E] | 50.20%[E] | 54.43%[E] | 51.75%[E] | 53.11%[E] | 76.55%[C] | 56.21%[E] | 85.67%[B] |
| 4 | 55.08%[E] | 50.37%[E] | 50.39%[E] | 52.03%[E] | 71.33%[C] | 58.16%[E] | 53.35%[E] | 53.08%[E] |
| 5 | 51.94%[E] | 50.30%[E] | 51.87%[E] | 66.74%[D] | 55.27%[E] | 51.22%[E] | 52.10%[E] | 51.40%[E] |
| 6 | 53.99%[E] | 52.39%[E] | 65.16%[D] | 51.65%[E] | 50.24%[E] | 50.37%[E] | 50.45%[E] | 55.84%[E] |
| 7 | 77.01%[C] | 72.78%[C] | 50.28%[E] | 50.64%[E] | 50.35%[E] | 50.38%[E] | 50.39%[E] | 52.31%[E] |
| 8 | 87.81%[B] | 84.20%[B] | 61.57%[D] | 54.46%[E] | 54.02%[E] | 54.68%[E] | 53.89%[E] | 56.30%[E] |
| 9 | 68.78%[D] | 54.11%[E] | 50.43%[E] | 50.38%[E] | 50.39%[E] | 53.62%[E] | 56.84%[E] | 73.32%[C] |
| 10 | 53.89%[E] | 50.47%[E] | 52.08%[E] | 54.45%[E] | 54.02%[E] | 50.46%[E] | 58.62%[E] | 51.28%[E] |
| 11 | 53.60%[E] | 51.72%[E] | 53.56%[E] | 52.75%[E] | 50.34%[E] | 52.72%[E] | 50.35%[E] | 53.65%[E] |
| 12 | 54.75%[E] | 50.31%[E] | 51.60%[E] | 50.34%[E] | 56.06%[E] | 50.37%[E] | 50.31%[E] | 51.30%[E] |
| 13 | 52.61%[E] | 50.44%[E] | 50.31%[E] | 53.20%[E] | 50.26%[E] | 52.22%[E] | 50.34%[E] | 52.71%[E] |
| 14 | 55.93%[E] | 51.64%[E] | 53.66%[E] | 50.47%[E] | 50.31%[E] | 50.40%[E] | 57.05%[E] | 60.47%[D] |
| 15 | 84.96%[B] | 65.46%[D] | 57.44%[E] | 57.70%[E] | 57.62%[E] | 67.07%[D] | 74.23%[C] | 85.81%[B] |

Similarly, the original SPECK32 rotation parameters (7,2) is not the optimal choice, and its accuracy is higher than multiple rotation parameters. The accuracy is the lowest using (3,9) as the rotation parameter. Table 5 shows that there are 60 rotation parameters with accuracy less than 51%. In the mode of choosing plaintext differences from optimal differential characteristics, (3,9) is the best choice. Considering of the computational error

**Table 6 The comparison table about using different kinds of plaintext difference.**

| $Acc_{(\alpha,\beta)}$ | $\beta$ | | | | | | | |
|---|---|---|---|---|---|---|---|---|
| | 0 | 1 | 2 | 3 | 4 | 5 | 6 | 7 |
| $\alpha$ | | | | | | | | |
| 0 | 0.00% | 0.08% | 0.37% | 1.28% | 8.71% | 15.63% | 14.05% | 20.58% |
| 1 | 4.74% | 1.01% | 20.83% | 7.07% | 6.38% | 9.71% | 8.39% | 0.26% |
| 2 | 18.98% | 15.66% | −7.62% | 0.18% | 0.24% | −2.44% | −0.81% | 7.62% |
| 3 | 27.59% | 15.29% | 0.05% | −2.03% | 0.19% | 0.19% | 0.36% | 0.73% |
| 4 | 24.31% | 7.38% | 0.27% | 0.36% | −0.60% | 0.19% | −0.48% | −0.93% |
| 5 | 14.64% | 8.52% | 0.10% | 0.18% | 0.23% | −1.99% | 4.65% | 0.39% |
| 6 | 14.95% | 5.81% | −0.17% | −1.01% | −2.62% | 3.36% | −1.80% | 0.20% |
| 7 | 13.39% | 5.40% | 4.16% | 0.14% | 0.22% | 0.30% | 0.20% | −3.16% |
| 8 | 0.15% | 0.16% | 1.64% | 0.43% | 0.56% | 2.06% | 2.11% | 9.96% |
| 9 | 19.46% | 6.85% | 0.07% | 0.06% | 1.40% | 0.12% | 9.50% | 10.97% |
| 10 | 18.99% | 8.34% | −0.22% | 0.25% | 0.19% | 0.64% | 0.21% | 5.94% |
| 11 | 15.75% | 3.61% | 0.21% | 0.33% | 3.94% | 2.02% | 4.99% | 0.15% |
| 12 | 15.87% | 9.32% | 8.88% | 10.66% | −0.27% | 4.37% | −0.49% | 0.28% |
| 13 | 10.04% | 21.12% | 2.09% | 2.07% | 6.23% | −6.61% | −3.26% | 0.15% |
| 14 | 3.21% | −0.04% | 0.59% | 16.48% | 9.59% | −0.69% | −0.80% | 0.27% |
| 15 | 1.29% | 1.07% | 4.49% | 12.90% | 3.93% | 0.17% | 2.31% | 3.51% |
| | 8 | 9 | 10 | 11 | 12 | 13 | 14 | 15 |
| 0 | −0.03% | 11.43% | 13.69% | 14.74% | 24.75% | 27.13% | 24.04% | 10.65% |
| 1 | 9.29% | 7.73% | 13.74% | 1.51% | 27.03% | −0.13% | 17.05% | 0.00% |
| 2 | 4.60% | 0.31% | −1.14% | −8.36% | 15.77% | 32.52% | 15.86% | 13.11% |
| 3 | 0.84% | 0.46% | −3.89% | −1.10% | 13.56% | 12.62% | 38.57% | 13.45% |
| 4 | −0.19% | 0.19% | 0.18% | 4.38% | 0.06% | 8.68% | 14.89% | 29.31% |
| 5 | 2.97% | 0.25% | 2.72% | 0.35% | 2.64% | −0.72% | −1.48% | 11.26% |
| 6 | 5.35% | 7.47% | 2.33% | 5.58% | 0.35% | 0.25% | 0.21% | 10.80% |
| 7 | −0.12% | 0.61% | 12.43% | −0.07% | 0.28% | 0.14% | 0.29% | 4.97% |
| 8 | 0.46% | −0.46% | 5.65% | 0.50% | 0.66% | 2.03% | 3.44% | 13.60% |
| 9 | 2.60% | −3.54% | 0.18% | 0.24% | 1.81% | −2.26% | 4.06% | 8.19% |
| 10 | 6.69% | 0.14% | −1.47% | 3.31% | 0.82% | 0.10% | 0.04% | 13.29% |
| 11 | 3.05% | −1.14% | 3.03% | 2.18% | 0.34% | −0.17% | 0.26% | 11.01% |
| 12 | 0.35% | 0.22% | −1.06% | 0.21% | −1.41% | 0.28% | 0.32% | 10.20% |
| 13 | 2.49% | 0.14% | 0.22% | 0.16% | 0.24% | −1.54% | 0.31% | 6.45% |
| 14 | 9.03% | 3.10% | −0.46% | 0.12% | 0.33% | 0.15% | −6.38% | 14.13% |
| 15 | 0.07% | 3.69% | 1.21% | 3.35% | 8.19% | 3.89% | 4.84% | 5.50% |

in calculating the accuracy, other rotation parameters, including (7,3), are also exemplary parameters, whose accuracy is less than 51%.

Considering the results in Tables 4 and 5, (7,3) is found to be a better choice with a lower accuracy than 50.5% using two kinds of plaintext differences. The SPECK32$_{(7,3)}$ has a stronger ability to resist distinguishing attacks based on neural distinguishers.

## Discussion

### A comparison using two kinds of plaintext differences

In Table 6, we give a comparison of using two types of plaintext differences to train $\mathcal{ND}$s. Let $Acc_{difference} = Acc_l - Acc_o$, where $Acc_l$ is the value shown in Table 4 and $Acc_o$ is the value shown in Table 5. As shown in Table 6, for most rotation parameters, the accuracy using low-Hamming differences is higher. For few rotation parameters, using input differences from optimal truncated differential characteristics to train neural distinguishers has more advantages over using low-Hamming-weight plaintext differences.

For using low-Hamming weight plaintext differences, there are more plaintext differences used to train $\mathcal{ND}$s. In contrast, there are few input differences obtained from optimal truncated differences, which is the main reason why the accuracy in Table 5 is lower for most rotation parameters. However, the methods that use input differences from optimal truncated differentials make sense, which helps designers and attackers to obtain better $\mathcal{ND}$ in some special cases like SPECK32$_{(13,5)}$.

For an attacker, it is better to use two types of plaintext difference model, if the attacker has enough time to train $\mathcal{ND}$s.

### A comparison using differential characteristics and neural distinguishers

*Kölbl, Leander & Tiessen (2015)* check all rotation parameters of SIMON for diffusion properties and optimal differential characteristics. Encouraged by their work, for each of the SPECK32-like ciphers, we calculate the probability of optimal seven-round differential characteristics, which is shown in Table 7.

As shown in Tables 4 and 7, the accuracy of neural distinguisher does not have a positive relationship with the differential probability. The differential probability of a rotation parameter with higher accuracy is not necessarily lower. For example, considering the rotation parameters (7,10) and (8,4), the differential probability of using (8,4) is higher than that of using (7,10). From the perspective of optimal differential probability, it is obvious that the SPECK32-like cipher using (8,4) as rotation parameters has better differential diffusion than using (7,10). But the accuracy of (7,10) is higher than the accuracy of (8,4); that is, the SPECK32-like cipher using (7,10) has a stronger ability to resist the distinguishing attack based on neural distinguishers.

*Benamira et al. (2021)* research into the phenomenon related to the accuracy and the differential probability. And they give an interpretation on why Gohr chooses $(0x40, 0x0)$ as the plaintext difference to train the five-round neural distinguisher instead of $(0x2800, 0x10)$, where $(0x2800, 0x10)$ is the input difference of the best five-round differential characteristic. They believe that this is explained by the fact that $(0x40, 0x0)$ is the input difference of the optimal three-round or four-round differential, which has the most chances to provide a biased distribution one or two rounds later.

With the existence of this phenomenon, it is necessary to evaluate the security using neural distinguishers, rather than relying solely on the optimal differential probabilities. The use of neural distinguishers in security evaluation enriches the results of security evaluation.

Table 7 The probabilities of optimal 7-round differential characteristics.

| $P_{(\alpha,\beta)}$ $\alpha$ \ $\beta$ | 0 | 1 | 2 | 3 | 4 | 5 | 6 | 7 |
|---|---|---|---|---|---|---|---|---|
| 0 | $2^{-7}$ | $2^{-5}$ | $2^{-7}$ | $2^{-7}$ | $2^{-7}$ | $2^{-7}$ | $2^{-7}$ | $2^{-7}$ |
| 1 | $2^{-6}$ | $2^{-10}$ | $2^{-15}$ | $2^{-14}$ | $2^{-13}$ | $2^{-14}$ | $2^{-15}$ | $2^{-11}$ |
| 2 | $2^{-7}$ | $2^{-15}$ | $2^{-16}$ | $2^{-20}$ | $2^{-19}$ | $2^{-20}$ | $2^{-16}$ | $2^{-18}$ |
| 3 | $2^{-7}$ | $2^{-17}$ | $2^{-20}$ | $2^{-16}$ | $2^{-21}$ | $2^{-16}$ | $2^{-20}$ | $2^{-19}$ |
| 4 | $2^{-7}$ | $2^{-18}$ | $2^{-18}$ | $2^{-21}$ | $2^{-9}$ | $2^{-21}$ | $2^{-17}$ | $2^{-19}$ |
| 5 | $2^{-7}$ | $2^{-18}$ | $2^{-20}$ | $2^{-16}$ | $2^{-21}$ | $2^{-14}$ | $2^{-16}$ | $2^{-20}$ |
| 6 | $2^{-7}$ | $2^{-17}$ | $2^{-16}$ | $2^{-17}$ | $2^{-18}$ | $2^{-16}$ | $2^{-16}$ | $2^{-20}$ |
| 7 | $2^{-7}$ | $2^{-14}$ | $2^{-18}$ | $2^{-21}$ | $2^{-21}$ | $2^{-19}$ | $2^{-20}$ | $2^{-16}$ |
| 8 | $2^{-4}$ | $2^{-11}$ | $2^{-13}$ | $2^{-13}$ | $2^{-10}$ | $2^{-13}$ | $2^{-13}$ | $2^{-13}$ |
| 9 | $2^{-7}$ | $2^{-16}$ | $2^{-20}$ | $2^{-20}$ | $2^{-19}$ | $2^{-21}$ | $2^{-18}$ | $2^{-11}$ |
| 10 | $2^{-7}$ | $2^{-17}$ | $2^{-19}$ | $2^{-18}$ | $2^{-19}$ | $2^{-17}$ | $2^{-11}$ | $2^{-18}$ |
| 11 | $2^{-7}$ | $2^{-17}$ | $2^{-20}$ | $2^{-17}$ | $2^{-18}$ | $2^{-11}$ | $2^{-18}$ | $2^{-21}$ |
| 12 | $2^{-7}$ | $2^{-18}$ | $2^{-17}$ | $2^{-18}$ | $2^{-8}$ | $2^{-18}$ | $2^{-18}$ | $2^{-19}$ |
| 13 | $2^{-7}$ | $2^{-17}$ | $2^{-16}$ | $2^{-11}$ | $2^{-19}$ | $2^{-17}$ | $2^{-18}$ | $2^{-20}$ |
| 14 | $2^{-7}$ | $2^{-12}$ | $2^{-11}$ | $2^{-17}$ | $2^{-18}$ | $2^{-20}$ | $2^{-19}$ | $2^{-20}$ |
| 15 | $2^{-5}$ | $2^{-10}$ | $2^{-15}$ | $2^{-18}$ | $2^{-18}$ | $2^{-18}$ | $2^{-17}$ | $2^{-17}$ |

| $\alpha$ \ $\beta$ | 8 | 9 | 10 | 11 | 12 | 13 | 14 | 15 |
|---|---|---|---|---|---|---|---|---|
| 0 | $2^{-4}$ | $2^{-7}$ | $2^{-7}$ | $2^{-7}$ | $2^{-7}$ | $2^{-7}$ | $2^{-7}$ | $2^{-6}$ |
| 1 | $2^{-10}$ | $2^{-15}$ | $2^{-15}$ | $2^{-14}$ | $2^{-13}$ | $2^{-13}$ | $2^{-11}$ | $2^{-8}$ |
| 2 | $2^{-13}$ | $2^{-19}$ | $2^{-19}$ | $2^{-18}$ | $2^{-18}$ | $2^{-15}$ | $2^{-11}$ | $2^{-12}$ |
| 3 | $2^{-13}$ | $2^{-18}$ | $2^{-18}$ | $2^{-19}$ | $2^{-19}$ | $2^{-11}$ | $2^{-14}$ | $2^{-13}$ |
| 4 | $2^{-11}$ | $2^{-21}$ | $2^{-18}$ | $2^{-19}$ | $2^{-8}$ | $2^{-19}$ | $2^{-17}$ | $2^{-14}$ |
| 5 | $2^{-13}$ | $2^{-21}$ | $2^{-17}$ | $2^{-11}$ | $2^{-19}$ | $2^{-19}$ | $2^{-17}$ | $2^{-14}$ |
| 6 | $2^{-13}$ | $2^{-18}$ | $2^{-11}$ | $2^{-17}$ | $2^{-19}$ | $2^{-18}$ | $2^{-19}$ | $2^{-15}$ |
| 7 | $2^{-11}$ | $2^{-11}$ | $2^{-19}$ | $2^{-21}$ | $2^{-21}$ | $2^{-18}$ | $2^{-20}$ | $2^{-15}$ |
| 8 | $2^{-4}$ | $2^{-11}$ | $2^{-13}$ | $2^{-13}$ | $2^{-10}$ | $2^{-13}$ | $2^{-13}$ | $2^{-12}$ |
| 9 | $2^{-13}$ | $2^{-16}$ | $2^{-20}$ | $2^{-20}$ | $2^{-19}$ | $2^{-18}$ | $2^{-18}$ | $2^{-11}$ |
| 10 | $2^{-13}$ | $2^{-19}$ | $2^{-16}$ | $2^{-15}$ | $2^{-18}$ | $2^{-20}$ | $2^{-16}$ | $2^{-15}$ |
| 11 | $2^{-13}$ | $2^{-18}$ | $2^{-16}$ | $2^{-14}$ | $2^{-21}$ | $2^{-16}$ | $2^{-20}$ | $2^{-14}$ |
| 12 | $2^{-11}$ | $2^{-21}$ | $2^{-17}$ | $2^{-21}$ | $2^{-9}$ | $2^{-21}$ | $2^{-18}$ | $2^{-14}$ |
| 13 | $2^{-13}$ | $2^{-21}$ | $2^{-18}$ | $2^{-16}$ | $2^{-21}$ | $2^{-16}$ | $2^{-21}$ | $2^{-14}$ |
| 14 | $2^{-13}$ | $2^{-18}$ | $2^{-16}$ | $2^{-20}$ | $2^{-19}$ | $2^{-20}$ | $2^{-16}$ | $2^{-14}$ |
| 15 | $2^{-11}$ | $2^{-15}$ | $2^{-18}$ | $2^{-18}$ | $2^{-18}$ | $2^{-18}$ | $2^{-15}$ | $2^{-10}$ |

# ANALYSIS OF THE REASON FOR THE DIFFERENCE IN ACCURACY

In this section, we further explore the differences in rotation parameters using the results in "Evaluate the Security of Different Rotation Parameters using Neural Distinguishers". We first select five rotation parameters with accuracies of 50–60%, 60–70%, 70–80%,

**Table 8 List of rotation parameters.**

| Target cipher | Rotation parameters | Plaintext difference | Accuracy |
|---|---|---|---|
| 7-round SPECK32$_{(\alpha,\beta)}$ | $(15, 1)$ | $(0x1000, 0x2000)$ | 97.33% |
| | $(1, 7)$ | $(0x1, 0x0)$ | 85.55% |
| | $(7, 8)$ | $(0x40, 0x8000)$ | 76.89% |
| | $(3, 12)$ | $(0x40, 0x8)$ | 66.67% |
| | $(8, 11)$ | $(0x80, 0x0)$ | 54.96% |

80–90% and 90–100%, respectively. The five rotation parameters with different accuracies and their plaintext differences are shown in Table 8. Using the five rotation parameters, we analyze the difference in the accuracies of the neural distinguishers caused by different rotation parameters, from the perspective of ciphertext pairs and truncated differences.

### The difference in ciphertext pairs

We focus on the bit biases of the output difference. To start, we perform the following experiment (Experiment A):

- **Stage 1.** Generate $2^{25}$ plaintext pairs using $\Delta$ as plaintext differences.
- **Stage 2.** Encrypt $2^{25}$ plaintext pairs using the seven-round SPECK32$_{(\alpha,\beta)}$.
- **Stage 3.** Calculate the output differences of $2^{25}$ ciphertext pairs.
- **Stage 4.** Count the number of output differences in which the value of the $j^{th}$ bit is 1, denoted by $n_j$.
- **Stage 5.** The bit bias of the $j^{th}$ bit is $\frac{n_j}{2^{25}} - 0.5$.

The bits biases of the five rotation parameters are shown in Table 9.

As shown in Table 9, for the rotation parameters (15,1), (1,7), (7,8), it is obvious that the partial bits have a probability of 0 (or 1) higher than 0.6. These bits with a probability of 0 (or 1) higher than 0.6 are denoted by good bits (**GB**s). For example, for the 29th bit of the rotation parameters (15,1) with $(0x1000, 0x2000)$ as the plaintext differences, it has a probability of 0 of about 0.911. For the 18th bit of the rotation parameters (1,7) with $(0x1, 0x0)$ as plaintext differences, it has a probability of 1 of approximately 0.810. Analyzing Table 9, (15,1) has more **GB**s than other rotation parameters. And (1,7) also has more **GB**s than (7,8). This phenomenon indicates that the higher the number of **GB**, the higher the accuracy of the neural distinguisher seems to have.

However, it is difficult to find the difference between (3,12) and (8,11) in the number of **GB**. So we further record the number of **GB** in truncated differences.

### The difference in truncated differences

For the seven-round SPECK32$_{(\alpha,\beta)}$, we focus on the bit biases of the truncated differences. With that, we conduct another experiment (Experiment B):

- **Stage 1.** Generate $2^{25}$ plaintext pairs using $\Delta$ as plaintext differences.
- **Stage 2.** Encrypt $2^{25}$ plaintext pairs using the seven-round SPECK32$_{(\alpha,\beta)}$.

**Table 9 Bits biases on output differences.**

| Rotation parameter (15,1) plaintext difference (0x1000, 0x2000) | Bit position | 31 | 30 | 29 | 28 | 27 | 26 | 25 | 24 |
|---|---|---|---|---|---|---|---|---|---|
| | | 0.012 | 0.024 | −0.411 | −0.377 | −0.334 | −0.283 | −0.224 | −0.161 |
| | Bit position | 23 | 22 | 21 | 20 | 19 | 18 | 17 | 16 |
| | | −0.100 | −0.048 | −0.011 | 0.002 | −0.002 | −0.001 | 0.002 | 0.006 |
| | Bit position | 15 | 14 | 13 | 12 | 11 | 10 | 9 | 8 |
| | | 0.012 | 0.024 | −0.411 | −0.378 | −0.335 | −0.283 | −0.224 | −0.162 |
| | Bit position | 7 | 6 | 5 | 4 | 3 | 2 | 1 | 0 |
| | | −0.102 | −0.050 | −0.019 | −0.006 | 0.003 | −0.001 | 0.002 | 0.005 |
| Rotation parameter (1,7) plaintext difference (0x1, 0x0) | Bit position | 31 | 30 | 29 | 28 | 27 | 26 | 25 | 24 |
| | | −0.058 | −0.016 | −0.003 | −0.003 | 0.000 | 0.000 | 0.146 | −0.089 |
| | Bit position | 23 | 22 | 21 | 20 | 19 | 18 | 17 | 16 |
| | | −0.038 | −0.010 | −0.005 | −0.004 | −0.000 | 0.310 | −0.282 | −0.213 |
| | Bit position | 15 | 14 | 13 | 12 | 11 | 10 | 9 | 8 |
| | | −0.041 | −0.009 | −0.004 | −0.008 | −0.014 | 0.000 | −0.157 | −0.104 |
| | Bit position | 7 | 6 | 5 | 4 | 3 | 2 | 1 | 0 |
| | | −0.059 | −0.010 | −0.004 | −0.000 | 0.001 | 0.000 | 0.228 | −0.174 |
| Rotation parameter (7,8) plaintext difference (0x40, 0x8000) | Bit position | 31 | 30 | 29 | 28 | 27 | 26 | 25 | 24 |
| | | −0.026 | −0.011 | −0.003 | −0.000 | 0.000 | 0.000 | 0.054 | −0.052 |
| | Bit position | 23 | 22 | 21 | 20 | 19 | 18 | 17 | 16 |
| | | −0.025 | −0.010 | −0.003 | −0.000 | 0.001 | −0.000 | 0.082 | −0.093 |
| | Bit position | 15 | 14 | 13 | 12 | 11 | 10 | 9 | 8 |
| | | −0.029 | −0.012 | −0.004 | −0.000 | 0.002 | −0.000 | −0.098 | −0.051 |
| | Bit position | 7 | 6 | 5 | 4 | 3 | 2 | 1 | 0 |
| | | −0.028 | −0.011 | −0.003 | −0.000 | 0.000 | −0.000 | −0.163 | −0.148 |
| Rotation parameter (3,12) plaintext difference (0x40, 0x8) | Bit position | 31 | 30 | 29 | 28 | 27 | 26 | 25 | 24 |
| | | −0.000 | 0.000 | 0.005 | 0.001 | −0.002 | −0.000 | 0.000 | −0.000 |
| | Bit position | 23 | 22 | 21 | 20 | 19 | 18 | 17 | 16 |
| | | −0.000 | 0.000 | 0.000 | −0.000 | 0.001 | −0.000 | −0.001 | 0.000 |
| | Bit position | 15 | 14 | 13 | 12 | 11 | 10 | 9 | 8 |
| | | −0.001 | 0.000 | 0.000 | 0.000 | −0.003 | −0.001 | −0.001 | 0.000 |
| | Bit position | 7 | 6 | 5 | 4 | 3 | 2 | 1 | 0 |
| | | −0.001 | −0.000 | 0.000 | −0.000 | −0.001 | −0.000 | 0.000 | 0.001 |
| Rotation parameter (8,11) plaintext difference (0x80, 0x0) | Bit position | 31 | 30 | 29 | 28 | 27 | 26 | 25 | 24 |
| | | 0.000 | 0.000 | −0.003 | 0.000 | 0.007 | −0.001 | 0.001 | −0.006 |
| | Bit position | 23 | 22 | 21 | 20 | 19 | 18 | 17 | 16 |
| | | −0.000 | 0.000 | −0.004 | −0.000 | −0.009 | −0.002 | −0.000 | 0.006 |
| | Bit position | 15 | 14 | 13 | 12 | 11 | 10 | 9 | 8 |
| | | 0.000 | −0.000 | −0.001 | 0.000 | −0.004 | −0.000 | −0.000 | 0.001 |
| | Bit position | 7 | 6 | 5 | 4 | 3 | 2 | 1 | 0 |
| | | 0.000 | 0.000 | −0.000 | −0.000 | −0.003 | −0.001 | 0.000 | −0.005 |

**Table 10 Bits biases on decrypting 1 round.**

| Rotation parameter (15,1) plaintext difference $(0x1000, 0x2000)$ | Bit position | 31 | 30 | 29 | 28 | 27 | 26 | 25 | 24 |
|---|---|---|---|---|---|---|---|---|---|
| | | 0.006 | 0.012 | −0.462 | −0.443 | −0.417 | −0.382 | −0.336 | −0.279 |
| | Bit position | 23 | 22 | 21 | 20 | 19 | 18 | 17 | 16 |
| | | −0.212 | −0.141 | −0.075 | −0.016 | 0.011 | −0.004 | 0.001 | 0.002 |
| | Bit position | 15 | 14 | 13 | 12 | 11 | 10 | 9 | 8 |
| | | 0.006 | 0.012 | −0.462 | −0.443 | −0.417 | −0.382 | −0.336 | −0.279 |
| | Bit position | 7 | 6 | 5 | 4 | 3 | 2 | 1 | 0 |
| | | −0.213 | −0.143 | −0.073 | −0.026 | −0.006 | 0.004 | 0.001 | −0.498 |
| Rotation parameter (1,7) plaintext difference $(0x1, 0x0)$ | Bit position | 31 | 30 | 29 | 28 | 27 | 26 | 25 | 24 |
| | | −0.069 | −0.013 | 0.000 | −0.000 | 0.000 | 0.245 | −0.177 | −0.109 |
| | Bit position | 23 | 22 | 21 | 20 | 19 | 18 | 17 | 16 |
| | | −0.048 | −0.014 | −0.016 | 0.000 | −0.409 | −0.377 | −0.352 | −0.266 |
| | Bit position | 15 | 14 | 13 | 12 | 11 | 10 | 9 | 8 |
| | | −0.048 | −0.009 | −0.003 | 0.000 | −0.001 | 0.255 | −0.191 | −0.128 |
| | Bit position | 7 | 6 | 5 | 4 | 3 | 2 | 1 | 0 |
| | | −0.069 | −0.010 | −0.014 | −0.053 | 0.000 | 0.332 | −0.282 | −0.213 |
| Rotation parameter (7,8) plaintext difference $(0x40, 0x8000)$ | Bit position | 31 | 30 | 29 | 28 | 27 | 26 | 25 | 24 |
| | | −0.088 | −0.045 | −0.017 | −0.004 | 0.000 | 0.001 | −0.000 | −0.148 |
| | Bit position | 23 | 22 | 21 | 20 | 19 | 18 | 17 | 16 |
| | | −0.092 | −0.048 | −0.020 | −0.007 | −0.000 | −0.001 | 0.000 | −0.221 |
| | Bit position | 15 | 14 | 13 | 12 | 11 | 10 | 9 | 8 |
| | | −0.092 | −0.049 | −0.021 | −0.006 | −0.001 | 0.000 | 0.115 | −0.148 |
| | Bit position | 7 | 6 | 5 | 4 | 3 | 2 | 1 | 0 |
| | | −0.095 | −0.052 | −0.023 | −0.008 | −0.002 | −0.000 | 0.155 | −0.290 |
| Rotation parameter (3,12) plaintext difference $(0x40, 0x8)$ | Bit position | 31 | 30 | 29 | 28 | 27 | 26 | 25 | 24 |
| | | −0.045 | −0.010 | −0.001 | 0.004 | −0.004 | −0.000 | 0.000 | 0.001 |
| | Bit position | 23 | 22 | 21 | 20 | 19 | 18 | 17 | 16 |
| | | 0.000 | 0.002 | 0.000 | 0.000 | −0.000 | −0.000 | 0.000 | 0.251 |
| | Bit position | 15 | 14 | 13 | 12 | 11 | 10 | 9 | 8 |
| | | −0.045 | −0.010 | −0.001 | 0.004 | −0.002 | −0.001 | −0.001 | 0.000 |
| | Bit position | 7 | 6 | 5 | 4 | 3 | 2 | 1 | 0 |
| | | −0.003 | −0.001 | −0.000 | −0.000 | −0.011 | −0.002 | −0.000 | 0.000 |
| Rotation parameter (8,11) plaintext difference $(0x80, 0x0)$ | bit position | 31 | 30 | 29 | 28 | 27 | 26 | 25 | 24 |
| | | −0.002 | 0.000 | 0.019 | 0.000 | 0.050 | −0.012 | 0.000 | −0.029 |
| | bit position | 23 | 22 | 21 | 20 | 19 | 18 | 17 | 16 |
| | | −0.007 | −0.000 | 0.027 | −0.018 | −0.044 | −0.020 | −0.002 | −0.043 |
| | Bit position | 15 | 14 | 13 | 12 | 11 | 10 | 9 | 8 |
| | | −0.002 | −0.000 | 0.006 | 0.000 | 0.023 | −0.002 | 0.000 | −0.013 |
| | Bit position | 7 | 6 | 5 | 4 | 3 | 2 | 1 | 0 |
| | | −0.003 | −0.000 | 0.024 | −0.016 | 0.031 | −0.011 | 0.000 | 0.029 |

**Table 11  Bits biases on decrypting 2 rounds.**

| Rotation parameter (15,1) plaintext difference $(0x1000, 0x2000)$ | Bit position | 31 | 30 | 29 | 28 | 27 | 26 | 25 | 24 |
| --- | --- | --- | --- | --- | --- | --- | --- | --- | --- |
| | | 0.002 | 0.005 | −0.487 | −0.480 | −0.468 | −0.450 | −0.425 | −0.389 |
| | Bit position | 23 | 22 | 21 | 20 | 19 | 18 | 17 | 16 |
| | | −0.339 | −0.275 | −0.198 | −0.114 | −0.030 | 0.008 | 0.000 | 0.001 |
| | Bit position | 15 | 14 | 13 | 12 | 11 | 10 | 9 | 8 |
| | | 0.002 | 0.005 | −0.487 | −0.480 | −0.468 | −0.451 | −0.425 | −0.389 |
| | Bit position | 7 | 6 | 5 | 4 | 3 | 2 | 1 | 0 |
| | | −0.339 | −0.276 | −0.196 | −0.122 | −0.039 | −0.008 | 0.000 | 0.000 |
| Rotation parameter (1,7) plaintext difference $(0x1, 0x0)$ | Bit position | 31 | 30 | 29 | 28 | 27 | 26 | 25 | 24 |
| | | −0.089 | −0.000 | −0.000 | −0.000 | 0.354 | −0.290 | −0.213 | −0.131 |
| | Bit position | 23 | 22 | 21 | 20 | 19 | 18 | 17 | 16 |
| | | −0.054 | 0.000 | −0.000 | −0.480 | −0.469 | −0.453 | −0.438 | −0.375 |
| | Bit position | 15 | 14 | 13 | 12 | 11 | 10 | 9 | 8 |
| | | −0.052 | −0.003 | −0.000 | −0.004 | 0.361 | −0.302 | −0.232 | −0.155 |
| | Bit position | 7 | 6 | 5 | 4 | 3 | 2 | 1 | 0 |
| | | −0.071 | −0.013 | −0.000 | −0.000 | −0.437 | −0.402 | −0.352 | −0.266 |
| Rotation parameter (7,8) plaintext difference $(0x40, 0x8000)$ | Bit position | 31 | 30 | 29 | 28 | 27 | 26 | 25 | 24 |
| | | −0.232 | −0.160 | −0.093 | −0.041 | −0.012 | −0.000 | 0.001 | −0.011 |
| | Bit position | 23 | 22 | 21 | 20 | 19 | 18 | 17 | 16 |
| | | −0.221 | −0.147 | −0.079 | −0.027 | 0.000 | 0.000 | 0.000 | −0.010 |
| | Bit position | 15 | 14 | 13 | 12 | 11 | 10 | 9 | 8 |
| | | −0.233 | −0.162 | −0.095 | −0.044 | −0.014 | 0.004 | −0.001 | −0.290 |
| | Bit position | 7 | 6 | 5 | 4 | 3 | 2 | 1 | 0 |
| | | −0.224 | −0.152 | −0.085 | −0.036 | −0.008 | 0.004 | 0.000 | −0.432 |
| Rotation parameter (3,12) plaintext difference $(0x40, 0x8)$ | Bit position | 31 | 30 | 29 | 28 | 27 | 26 | 25 | 24 |
| | | 0.075 | −0.023 | 0.005 | 0.002 | 0.020 | 0.004 | 0.002 | 0.000 |
| | Bit position | 23 | 22 | 21 | 20 | 19 | 18 | 17 | 16 |
| | | 0.000 | −0.000 | 0.000 | 0.002 | 0.250 | −0.250 | −0.250 | −0.247 |
| | Bit position | 15 | 14 | 13 | 12 | 11 | 10 | 9 | 8 |
| | | −0.075 | −0.022 | 0.005 | 0.002 | −0.022 | −0.002 | −0.001 | 0.001 |
| | Bit position | 7 | 6 | 5 | 4 | 3 | 2 | 1 | 0 |
| | | −0.029 | −0.008 | −0.000 | −0.000 | −0.251 | −0.250 | −0.250 | −0.262 |
| Rotation parameter (8,11) plaintext difference $(0x80, 0x0)$ | Bit position | 31 | 30 | 29 | 28 | 27 | 26 | 25 | 24 |
| | | −0.041 | −0.000 | −0.077 | −0.041 | −0.001 | −0.097 | −0.070 | −0.171 |
| | Bit position | 23 | 22 | 21 | 20 | 19 | 18 | 17 | 16 |
| | | −0.068 | −0.001 | 0.063 | −0.027 | 0.000 | −0.068 | 0.000 | 0.276 |
| | Bit position | 15 | 14 | 13 | 12 | 11 | 10 | 9 | 8 |
| | | −0.008 | −0.000 | −0.057 | −0.032 | −0.000 | −0.090 | −0.064 | 0.107 |
| | Bit position | 7 | 6 | 5 | 4 | 3 | 2 | 1 | 0 |
| | | −0.034 | 0.000 | 0.052 | −0.025 | −0.000 | −0.023 | 0.000 | −0.076 |

**Table 12 The number of *GB*s.**

| Target cipher | Rotation parameters | Plaintext difference | Accuracy | Output difference | Decrypt 1 round | Decrypt 2 rounds |
|---|---|---|---|---|---|---|
| 7-round SPECK32$_{(\alpha,\beta)}$ | (15,1) | $(0x1000, 0x2000)$ | 97.33% | 14 | 17 | 20 |
| | (1,7) | $(0x1, 0x0)$ | 85.55% | 8 | 13 | 17 |
| | (7,8) | $(0x40, 0x8000)$ | 76.89% | 2 | 6 | 10 |
| | (3,12) | $(0x40, 0x8)$ | 66.67% | 0 | 1 | 8 |
| | (8,11) | $(0x80, 0x0)$ | 54.96% | 0 | 0 | 3 |

- **Stage 3.** For the $2^{25}$ cipher pairs, decrypt $i$ rounds using their respective keys.
- **Stage 4.** Compute the corresponding truncated differences.
- **Stage 5.** Compute the bit bias of the truncated differences.

The bit biases of the five rotation parameters are shown in Tables 10 and 11. Similarly, (15,1) has more *GB*s than other rotation parameters in truncated differences. The number of *GB* is shown in Table 12. It is found that there is a positive correlation between the accuracy of the neural distinguishers and the number of *GB*, which also proves that the neural network needs more *GB*s. More *GB*s, the neural distinguisher appears to have higher accuracy.

## CONCLUSION AND FUTURE WORK

In this work, we present a comprehensive security assessment of SPECK32 variants with different rotation parameters considering the ability to resist neural-distinguishing attack. First, we train neural distinguishers for all SPECK32-like ciphers using two distinct plaintext difference selection strategies. Subsequently, employing neural distinguisher accuracy as our primary evaluation metric, we conduct a rigorous security assessment of all rotation parameter configurations.

In particular, our experimental results reveal that the standard parameter (7,2) does not consistently demonstrate optimal security characteristics. Through comparative analysis, we identify the parameter configuration (7,3) as exhibiting superior resistance against neural distinguisher-based distinguishing attacks compared to the standard parameter. This finding suggests potential security considerations for parameter selection in SPECK32-like ciphers.

Furthermore, we establish through empirical validation a significant correlation between ciphertext pair bit biases and neural distinguisher accuracy, particularly in truncated-round scenarios. Our analysis provides new insights into the interpret ability of neural cryptanalysis.

Our work is the first time to evaluate the security of the rotation parameters of SPECK32-like ciphers using neural distinguishers. The use of neural distinguishers enriches the results of the security evaluation of cryptographic components.

Although machine learning demonstrates significant potential for cryptanalysis, we do not think that machine learning methods will replace traditional cryptanalysis. Serving as a powerful complement to classical cryptanalysis, machine learning-aided methods enable

researchers to identify previously unnoticed vulnerabilities. In further work, an interesting direction is to utilize the weakness found by the neural distinguishers to enhance classical cryptanalysis.

### Funding

This work was supported by the National Natural Science Foundation of China (No. 62206312). The funders had no role in study design, data collection and analysis, decision to publish, or preparation of the manuscript.

### Grant Disclosures

The following grant information was disclosed by the authors:
National Natural Science Foundation of China: 62206312.

### Competing Interests

The authors declare that they have no competing interests.

### Author Contributions

- Zezhou Hou conceived and designed the experiments, performed the experiments, analyzed the data, performed the computation work, prepared figures and/or tables, authored or reviewed drafts of the article, and approved the final draft.
- Jiongjiong Ren conceived and designed the experiments, analyzed the data, performed the computation work, prepared figures and/or tables, authored or reviewed drafts of the article, and approved the final draft.
- Shaozhen Chen analyzed the data, performed the computation work, authored or reviewed drafts of the article, and approved the final draft.

### Data Availability

   The code is available at Zenodo: free crypto. (2025). ml-aided-crypto/speck32_security_ml_analysis: speck32_like_al_for_cryptanalysis (v1.0). Zenodo. https://doi.org/10.5281/zenodo.15624749.

### Supplemental Information

Supplemental information for this article can be found online at http://dx.doi.org/10.7717/peerj-cs.3015#supplemental-information.

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
