# Peer review of "Further observations on the security of Speck32-like ciphers using machine learning"

_PeerJ Computer Science, doi:10.7717/peerj-cs.3015_

## Round 0.1 · original submission · Minor Revisions

The referral process is now complete. While finding your paper interesting, the referees and I feel that more work could be done before the paper is published. My decision is therefore to provisionally accept your paper subject to minor revisions.

Reviewer 1 ·

Basic reporting

The paper demonstrates clear and unambiguous use of professional English throughout, contributing to its readability and overall quality, but a couple of small adjustments would make it even better:

Line 54: The phrase "e.g." should not be followed directly by parentheses. It is better to use "including those by…" or "such as those by .."

Experimental design

The motivation for studying this cipher and the gap it addresses in the literature are clearly articulated. The authors effectively highlight the lack of parameter analysis for SPECK compared to SIMON-like ciphers, which justifies the relevance and necessity of their work.

Validity of the findings

The analysis of how different rotation parameters impact the accuracy of neural distinguishers, particularly in relation to ciphertext pairs and truncated differences, is insightful and well-explained.

Reviewer 2 ·

Basic reporting

The paper is well organized and clear. The results are interesting and explained in details.

Experimental design

Research question is interesting and well defined. Methods are described with details.

Validity of the findings

In this work, the authors present a security analysis of SPECK32 variants with different rotation parameters. The finding are explained in details.

However, the authors should include the paper "Machine Learning Attacks On SPECK" Anubhab Baksi, Jakub Breier, Vishnu Asutosh Dasu, Xiaolu Hou, Security and Implementation of Lightweight Cryptography (SILC), in the literature review part and compare the methods and results of the mentioned paper and their work.

Similarly, there is another recent paper: "New Results on Machine Learning-Based Distinguishers", Anubhab Baksi; Jakub Breier; Vishnu Asutosh Dasu; Xiaolu Hou; Hyunji Kim; Hwajeong Seo, IEEE Access. In this paper the authors presented distinguishers on up to 8-round SPECK-32 and 7-round SPECK-128, using MLPs.

In the contribution part the authors stated:
"Considering that the low-hamming weight input difference usually leads to a better differential characteristic, we train neural distinguishers of 7-round SPECK32-like using plaintext differences with hamming weight at most 2." The authors should compare their work and their results with "New Results on Machine Learning-Based Distinguishers".

Reviewer 3 ·

Basic reporting

The language is clear, unambiguous and fluent. Every part of the paper can be understood easily. The references seem to be proper and adequate. The article structure, figure and table enumeration are consistent. The results are explained clearly.

Experimental design

The study seems to follow scientific methods appropriately. It questions the values for the rotation parameters of Speck32 cipher. The methods are explained adequately and allowing to replay the tests.

Validity of the findings

The result seems to be accurate and meaningful.

---

## Round 0.2 · accepted · Accept

Since the comments have been addressed, we are happy to inform you that your manuscript has been accepted for publication.